# Regulated spindle orientation buffers tissue growth in the epidermis

Angel Morrow[1†‡], Julie Underwood[1†], Lindsey Seldin[1,2§], Taylor Hinnant[1,2], Terry Lechler[1,2]*

[1]Department of Dermatology, Duke University, Durham, United States; [2]Department of Cell Biology, Duke University, Durham, United States

**Abstract** Tissue homeostasis requires a balance between progenitor cell proliferation and loss. Mechanisms that maintain this robust balance are needed to avoid tissue loss or overgrowth. Here we demonstrate that regulation of spindle orientation/asymmetric cell divisions is one mechanism that is used to buffer changes in proliferation and tissue turnover in mammalian skin. Genetic and pharmacologic experiments demonstrate that asymmetric cell divisions were increased in hyperproliferative conditions and decreased under hypoproliferative conditions. Further, active K-Ras also increased the frequency of asymmetric cell divisions. Disruption of spindle orientation in combination with constitutively active K-Ras resulted in massive tissue overgrowth. Together, these data highlight the essential roles of spindle orientation in buffering tissue homeostasis in response to perturbations.
DOI: https://doi.org/10.7554/eLife.48482.001

**\*For correspondence:**
terry.lechler@duke.edu

[†]These authors contributed equally to this work

**Present address:** [‡]Novozymes, Durham, United Kingdom; [§]Department of Cell and Developmental Biology, Vanderbilt University, Nashville, United States

**Competing interests:** The authors declare that no competing interests exist.

## Introduction

The epidermis is a proliferative tissue that turns over repeatedly throughout life. Remarkably, in aged sun-exposed skin, homeostasis and the architecture of the epidermis is maintained despite prevalent clones of cells containing oncogenic mutations (*Martincorena et al., 2015*). Recent work has highlighted oncogene-induced differentiation and active elimination of mutant clones as two mechanisms of epidermal robustness, suggesting there are multiple modes of response to perturbation (*Brown et al., 2017*; *Ying et al., 2018*). Homeostasis requires that gain of cells (proliferation) must be balanced with loss of cells (differentiation or cell death). One possible mechanism to control homeostasis is regulated spindle orientation and asymmetric cell divisions. These divisions do not increase progenitor number, but rather commit one daughter to differentiation. Embryonic epidermal progenitors can divide parallel to the basement membrane, resulting in two basal daughter cells, which we term symmetric divisions. These cells can also divide perpendicular to the basement membrane (*Lechler and Fuchs, 2005*; *Poulson and Lechler, 2010*; *Smart, 1970*; *Williams et al., 2011*). Perpendicular divisions are termed asymmetric divisions because they contribute one daughter cell to the differentiated suprabasal cell layer and do not alter the number of basal progenitor cells. Because of these properties, regulated spindle orientation is a candidate for providing resilience to homeostatic perturbation. In the embryonic epidermis, asymmetric divisions are used to drive stratification of the skin (*Lechler and Fuchs, 2005*; *Smart, 1970*; *Williams et al., 2011*). However, in adult backskin most divisions are planar (symmetric) and the utility of regulated spindle orientation has not been explored (*Ipponjima et al., 2016*; *Rompolas et al., 2016*). Here, we examine the possibility that regulated spindle orientation can buffer the effects of perturbed homeostasis.

## Results

We began by examining whether spindle orientation was altered when developmental or adult homeostatic proliferation was experimentally altered. In the embryonic mouse epidermis, where proliferation is high, the ratio of perpendicular to planar divisions is approximately 70:30 (*Lechler and Fuchs, 2005*; *Smart, 1970*). To inhibit basal cell cycle progression, we used a basal epidermal keratin 14 promoter to drive a doxycycline-inducible allele of Cdkn1b, a CDK1 inhibitor (K14-rtTA;tetO-CDKN1b) (*Pruitt et al., 2013*). Quantitation of the percentage of cells expressing the mitotic marker phospho-histone H3 (pHH3) revealed a significant decrease in basal cell proliferation (*Figure 1A*). This also resulted in a significant reduction in perpendicular divisions, from 70% to 35%, with a concomitant increase in planar divisions (from 22% to 42%) (*Figure 1B,C*). We measured spindle angles as shown in *Figure 1—figure supplement 1*. This data is consistent with the need for increased rates of planar divisions to maintain the surface area of the epidermis in a rapidly growing embryo.

In contrast to the embryo, the proliferation rate in the adult backskin is low. Live imaging has shown that most divisions occur symmetrically and that basal cells delaminate from the basement membrane to populate suprabasal cell layers (*Ipponjima et al., 2016*; *Rompolas et al., 2016*). Consistent with this observation, we see that the majority of mitotic spindles are planar in adult backskin (*Figure 1E*). Notably, there is regional variability in epidermis from different anatomical areas in both proliferation and spindle orientation. The epidermis of the footpad, which is thicker, has both a

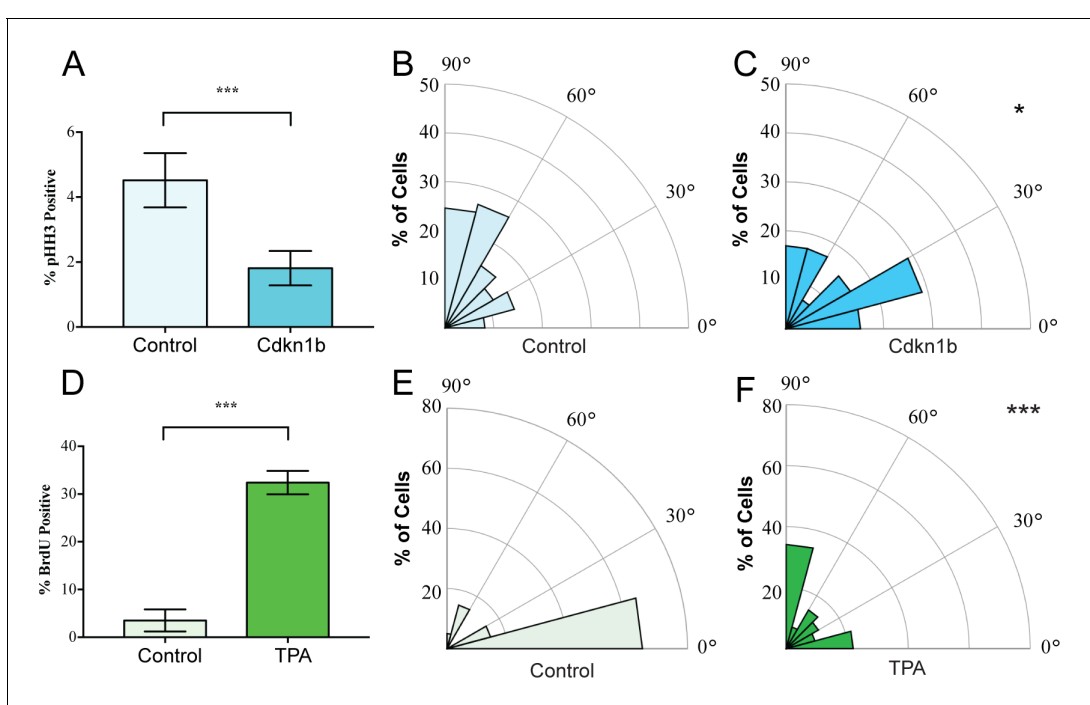

**Figure 1.** Mitotic spindles reorient in response to changes in proliferation. (**A**) Proliferation of control and K14-rtTA;TRE-Cdkn1b embryonic epidermis measured by pHH3 incorporation after treatment with doxycycline from e14.5–16.5. (**B–C**) Radial histograms of mitotic spindle orientation for e16.5 control (n = 61) and Cdkn1b (n = 59) embryonic epidermis. (**D**) Proliferation of adult backskin epidermis, control or TPA-treated (10 µl of 0.4 mM TPA, applied daily for 5 days), as measured by BrdU incorporation. n > 150 for each of three mice. (**E–F**) Radial histograms of mitotic spindle orientation for control (n = 20) and TPA-treated (n = 41) epidermis. Three or more mice were examined for each condition. *p<0.05, ***p<0.001.

DOI: https://doi.org/10.7554/eLife.48482.002

The following figure supplements are available for figure 1:

**Figure supplement 1.** Image of a dividing basal epidermal cell demonstrating the method of spindle angle measurement.
DOI: https://doi.org/10.7554/eLife.48482.003
**Figure supplement 2.** Proliferation and spindle orientation differences between back and footpad skin.
DOI: https://doi.org/10.7554/eLife.48482.004
**Figure supplement 3.** Expansion of keratin 10 (K10) positive cell layers in epidermis treated with TPA.
DOI: https://doi.org/10.7554/eLife.48482.005

higher proliferation index and increased numbers of perpendicular divisions (*Figure 1—figure supplement 2*). Similar to the embryonic epidermis, expression of Cdkn1b caused a shift to more planar cell divisions in the paw epidermis (*Figure 1—figure supplement 2E*).

To induce hyperproliferation of adult backskin epidermis we topically applied the mitogen TPA. This resulted in thickening of the epidermis and a 6-fold increase in proliferation rate as measured by BrdU incorporation (*Figure 1D*). Notably, this treatment increased the percentage of perpendicular divisions from 20% to 40% (*Figure 1E,F*). Therefore, one of the responses to TPA-induced hyperproliferation in the epidermis is an increase in the rate of divisions that give rise to differentiated progeny, as seen by staining for the differentiation marker, keratin 10 (*Figure 1—figure supplement 3*). Together these data demonstrate correlations between proliferation and spindle orientation, and support the hypothesis that cells tune their division orientation in response to the needs of the surrounding tissue as a mechanism to promote robust homeostasis.

To directly determine whether regulated spindle orientation was important for epidermal homeostasis we examined the effects of mutating the nuclear mitotic apparatus protein, NuMA, which is essential for spindle orientation in embryonic skin (*Williams et al., 2011*; *Seldin et al., 2016*). We did not see significant phenotypes when we disrupted NuMA's spindle orientation function in adult epidermis for short times (data not shown), consistent with findings that adult backskin homeostasis is normally maintained by delamination and differentiation rather than by regulated spindle orientation (*Rompolas et al., 2016*).

Given that tissues responded to TPA-induced hyperproliferation by increasing perpendicular divisions, we next asked whether this also occurred in response to an oncogenic perturbation. Mutations that activate Ras family members can drive squamous cell carcinoma and a significant percentage of squamous tumors carry Ras mutations (*Daya-Grosjean et al., 1993*; *Pierceall et al., 1991*; *South et al., 2014*). Unexpectedly, we found that expression of KRAS$^{G12D}$ (at endogenous levels and only in the epidermis; K5CreER;KRAS$^{G12D/+}$) had distinct effects in backskin and footpad epidermis. While there was no significant change in division orientation ratios in the backskin, active K-Ras resulted in an increase in perpendicular divisions in the footpads (*Figure 2A–D*). These data rule out the possibility that active K-Ras inhibits regulated spindle orientation/asymmetric cell divisions as an oncogenic mechanism. It raised the alternative possibility that the increased perpendicular divisions may protect the tissue against overgrowth. We therefore addressed the molecular requirements for spindle orientation in skin expressing active K-Ras. We combined a NuMA mutation that disrupts embryonic spindle orientation with the KRAS$^{G12D}$ oncogene (*Seldin et al., 2016*; *Silk et al., 2009*). This resulted in a significant shift of divisions from perpendicular to planar, consistent with NuMA being required for spindle orientation in this context (*Figure 2E*). On their own, mutations in NuMA resulted in a skewing of spindle orientations in the pawskin (*Figure 2—figure supplement 1*).

The consequence of perturbing regulated spindle orientation in epidermis expressing active KRAS was dramatic. Within two weeks of recombination, large overgrowths appeared on the footpads and anogenital regions of the double mutant mice (K5Cre$^{ER}$;KRAS$^{G12D/+}$;NuMA$^{MTBDfl/fl}$) (*Figure 3A*). These features were not found in single KRAS$^{G12D}$ or NuMA mutants, although KRAS$^{G12D}$ alone was sufficient for formation of oral tumors similar to those that appeared in the double mutants. Most mice died within 3–4 weeks after recombination (*Figure 3B*).

Histologic analysis of the double mutant paws demonstrated massive tissue overgrowth and papilloma formation (*Figure 3D–G*). Despite this dramatic change in tissue architecture, the basement membrane remained intact and differentiation was normal at early time points. Thus, this combination of mutations is not sufficient for invasion, at least in the short time period we were able to assay. The massive tissue overgrowth led to lethality, likely due, at least in part, to blockage of the oral cavity.

Similar to the loss of NuMA function in the adult epidermis, loss of spindle orientation alone is not detrimental in developing fly wings (*Nakajima et al., 2013*). However, tumors form when combined with additional mutations that inhibited apoptosis. Notably, we saw no genetic interaction when mutant p53 was combined with the NuMA$^{ΔMTBD}$ deletion (*Figure 3C*). These data are consistent with the fact that apoptosis is not a major determinant of tissue homeostasis in the epidermis under normal conditions, and demonstrates a specificity for disrupted spindle organization working synergistically with only a subset of oncogenic drivers.

Histologic analysis revealed expansion of the stem cell pool in the KRAS$^{G12D}$; NuMA$^{ΔMTBD}$ epidermis. To accommodate the increase in cell numbers, the basement membrane became highly

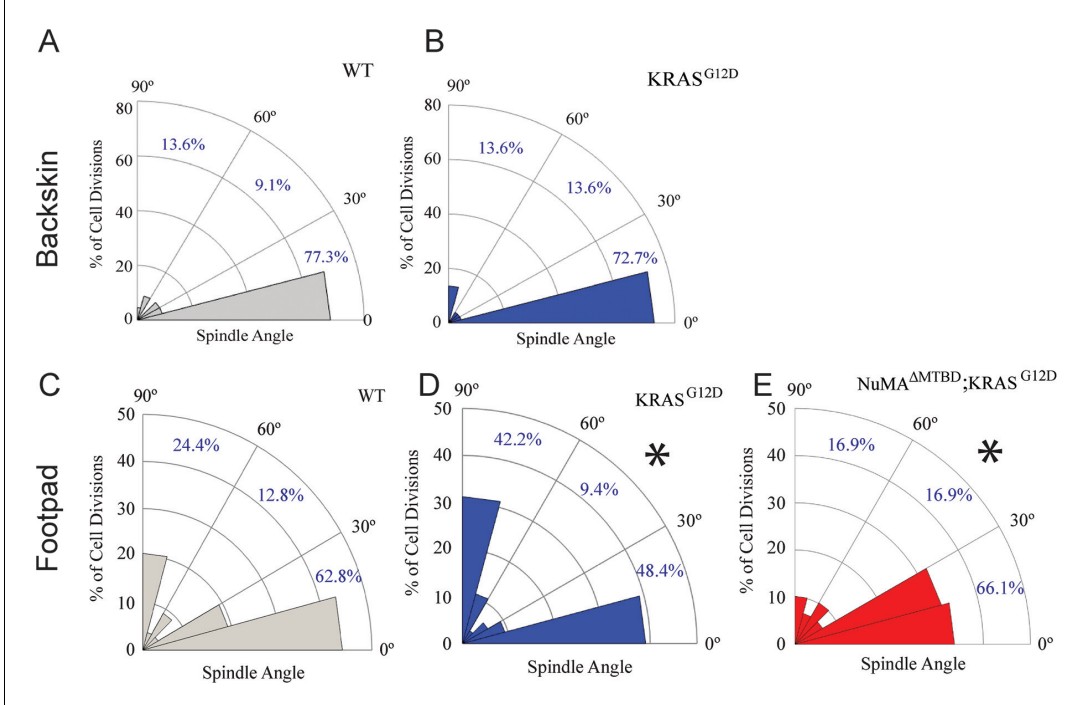

**Figure 2.** Oncogenic KRAS alters footpad epidermis spindle orientation in a NuMA-dependent manner. (A,B) Radial histogram of mitotic spindles in adult backskin 21 days after tamoxifen-induced recombination in control (A) and *K5Cre^ER; KRAS^G12D* mice (B). (C,D) Radial histogram of mitotic spindles in adult footpad epidermis, 21 days after tamoxifen-induced recombination in control (C) and *K5Cre^ER; KRAS^G12D* mice, (n = 78 cells), and (D) (n = 64 cells). (E) Radial histogram of mitotic spindles in adult footpad 21 days after tamoxifen-induced recombination in *K5Cre^ER; KRAS^G12D;NuMA^ΔMTBD* mice (n = 57 cells). Note that the data in 2C is the same as that presented in *Figure 1—figure supplement 2E*.

DOI: https://doi.org/10.7554/eLife.48482.006

The following figure supplement is available for figure 2:

**Figure supplement 1.** Radial histogram of cell division orientation in the footpad epidermis from NuMA mutant mice (K14-Cre;NuMA^MTBD/MTBD).

DOI: https://doi.org/10.7554/eLife.48482.007

involuted (*Figure 3D–G*). We quantitated this as basement membrane length per tissue length and found a 2–4 fold increase in this measure, corresponding to a 4–16 fold increase in area (*Figure 3H*). As cell density was not notably altered, this demonstrates an expansion of the basal stem cell pool. These data are consistent with regulated spindle orientation buffering the effects of oncogenic K-Ras. Similarly, we found that low-dose TPA treatment caused a subtle yet significant increase in basement membrane length in NuMA^ΔMTBD mice as compared to control mice (*Figure 3—figure supplement 1*).

Active K-Ras has both proliferative and anti-differentiative effects in different tissues. Because increased proliferation can drive increased asymmetric divisions (*Figure 1*), we asked whether the changes observed in K-Ras mutant skin could be due to oncogene-induced hyperproliferation. Notably, we found that constitutively active KRas expression is not sufficient to drive hyperproliferation in the paw skin, nor did it do so in combination with a mutation in NuMA (*Figure 3I*).

Previous work has suggested that active Ras can inhibit differentiation when overexpressed in the epidermis (*Dajee et al., 2002*). However, this has not been examined either in the adult pawskin or under endogenous levels of expression. In adults, cells commit to differentiation, detach from the basement membrane, and move upward. A small proportion of basal cells express the differentiation marker keratin 10 (K10) and can be fluorescently labeled in mice using established mouse lines (K10-rtTA;TRE-H2B-GFP) (*Muroyama and Lechler, 2017*; *Figure 3J*). In the pawskin, we found that ~10% of basal cells were labeled with H2B-GFP, and this fell to 0.4% two days after removal of doxycycline, consistent with these cells entering differentiation (n>200 cells). Additionally, H2B-GFP labeled-basal cells incorporated EdU at a lower rate than surrounding basal cells (28 ± 2.4% for K10^-ve basal cells verses 9 ± 3.0% for K10^+ve basal cells, n=3 mice, >400 cells/mouse). We quantitated

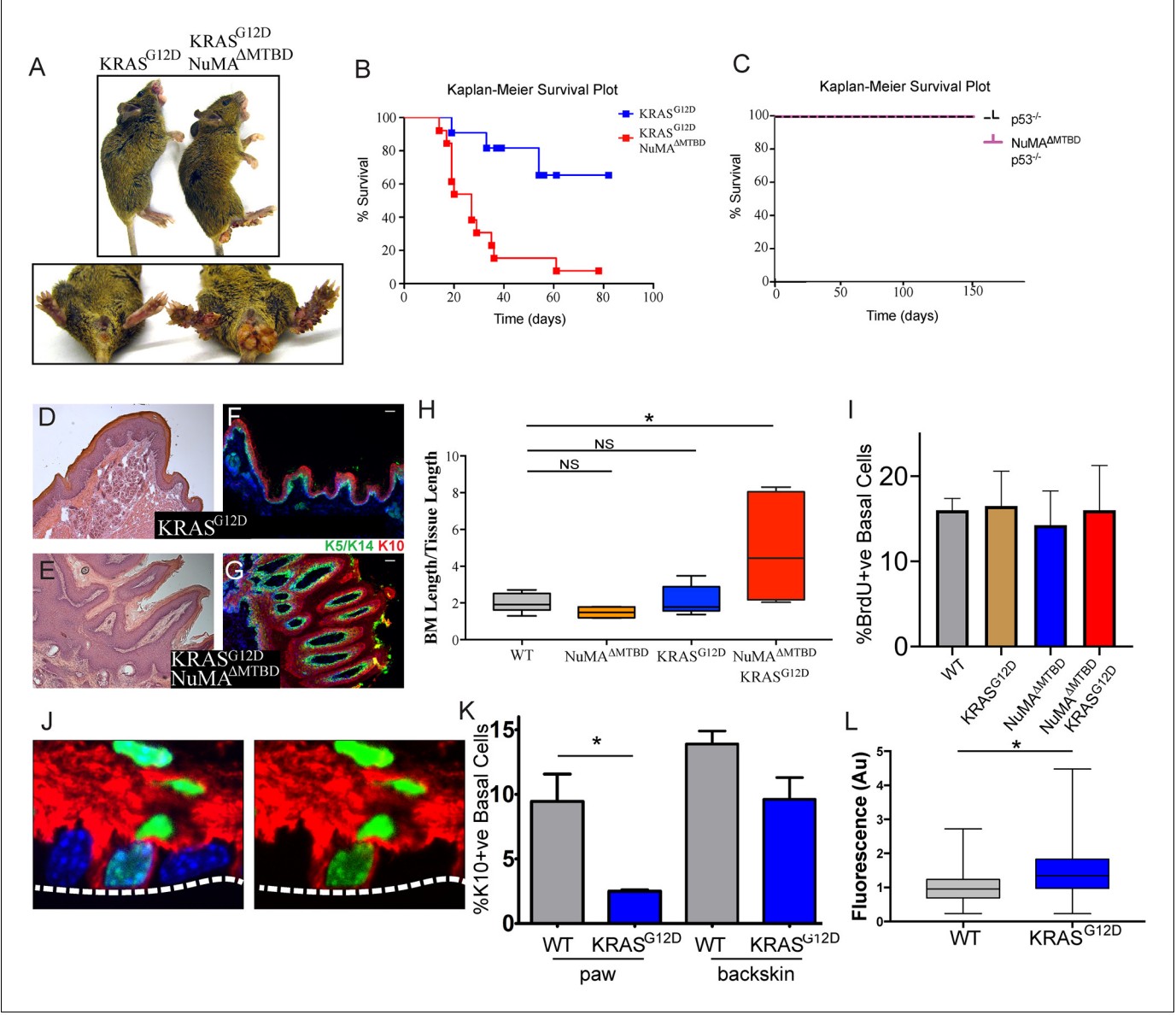

**Figure 3.** Loss of regulated spindle orientation synergizes with oncogenic KRAS to cause tissue overgrowth. (A) Images of *K5CreER; KRASG12D* and *K5CreER; NuMAΔMTBD; KRASG12D* mice 21 days after recombination with tamoxifen, with inset of footpad and anal-genital region. (B,C)) Kaplan-Meier Survival plot of *K5CreER; KRASG12D* and *K5CreER; NuMAΔMTBD; KRASG12D* mice (B) and *K5CreER; p53-/-* and *K5CreER; p53-/-; NuMAΔMTBD* mice (C). n = 12 mice for *K5CreER; NuMAΔMTBD; KRASG12D* and 10 for other genotypes. (D,E) H and E images of *K5CreER; KRASG12D* (D) and *K5CreER; NuMAΔMTBD; KRASG12D* (E) footpad epidermis. (F,G) Immunofluorescence images of *K5CreER; KRASG12D* (F) and *K5CreER; NuMAΔMTBD; KRASG12D* (G) footpad epidermis showing localization of K5/K14+ basal and K10+ suprabasal epidermal layers. Scale bar = 50 μm. (H) Quantitation of basement membrane length divided by tissue length in indicated genotypes (n = 3 mice/condition). (I) Quantitation of proliferation, as assayed by BrdU incorporation, in control and KRASG12D mice. (n > 300 cells, three mice/condition). (J) Image showing co-localization of keratin 10 (red) and histone H2B (green) in a basal cell from a *K10-rtTA; TRE-H2B-GFP* mouse. (K) Quantitation of Keratin 10 positive basal cells in paw and backskin with indicated genotypes. n > 300 cells/mouse, three mice/condition. (L) Fluorescence intensity (normalized) of β4-integrin in control and KRASG12D expressing footpad epidermis (n = 3 mice/condition).

DOI: https://doi.org/10.7554/eLife.48482.008

The following figure supplement is available for figure 3:

**Figure supplement 1.** Effect of TPA treatment on control and NuMAΔMTBD ear skin.

DOI: https://doi.org/10.7554/eLife.48482.009

differentiation by analyzing the percentage of basal cells that express keratin 10. Using the K10-rtTA;TRE-H2B-GFP genetic labeling system we found that differentiation was suppressed in KRAS$^{G12D}$ expressing cells (*Figure 3K*). This was most dramatic in the paw epidermis, but was also observed in backskin to a lesser extent. While the mechanism underlying this suppression is not clear, we also found that the relative intensity of β4-integrin was increased in the mutant pawskin, in a similar manner to what was observed when K-Ras was overexpressed in the embryo (*Dajee et al., 2002*; *Figure 3J*). Robust adhesion and signaling through integrins is known to promote progenitor cell fate, and thus may contribute to the changes in differentiation that we observed.

## Discussion

Together, our data demonstrate that the epidermis can respond to changes in homeostasis by altering mitotic spindle orientation. Increasing perpendicular (asymmetric) divisions in response to either increased proliferation or decreased differentiation allows homeostasis to be reset without dramatically affecting tissue architecture. Our genetic analysis clearly shows that this is essential to prevent tissue overgrowth in an oncogenic background. In addition, we found that the epidermis responded differently to the same oncogenic insult at different anatomic sites. This likely reflects the differences in proliferation, differentiation, and rates of asymmetric cell division rates in different regions of the epidermis and suggests that distinct 'second hits' may be needed to collaborate with oncogenes in tissues with different mechanisms of achieving homeostasis.

Our data are consistent with the idea that both regulated spindle orientation and delamination/differentiation can be used to buffer basal cell number. Disruption of either one was insufficient to induce homeostatic defects due to compensation by the other. However, blockade of both homeostatic buffers resulted in tissue overgrowth and progenitor cell expansion. Notably, squamous cell carcinomas have been reported to have increased rates of perpendicular divisions as compared to normal epidermis (*Beck et al., 2011*; *Siegle et al., 2014*). It will be important to see whether regulated spindle orientation is protective against tumor development in these contexts.

## Materials and methods

### Mouse strains and tissue preparation

All animal work was approved by Duke University's Institutional Animal Care and Use Committee. Mouse strains used in this study were: *Krt5*-Cre$^{ER}$ (Gift from Brigid Hogan), *NuMA1*-ΔMTBD (*Seldin et al., 2016*; *Silk et al., 2009*), *KRAS*$^{G12D}$ (*Jackson et al., 2001*), and *Trp53*$^{fl/fl}$ (*Marino et al., 2000*). Mice were genotyped by PCR, and both males and females were used for experiments from postnatal day 30–60. For BrdU incorporation experiments, mice were injected with 10 mg/kg BrdU (Sigma-Aldrich) and left for 75 mins prior to sacrifice. Paws and backskin were embedded in Optimal Cutting Temperature prior to freezing, and samples were stored at −80 ˚C. A cryostat was used to cut 10 μm thick sections. For TPA (Cayman Chemical) treatment, 10 μl of 0.4 mM TPA was applied to the backskin each day for 5 days.

### Immunofluorescence

For immunofluorescent staining, 10 uM thick sections were fixed for 8 min in 4% PFA in PBS-T (containing 0.2% Triton X-100) or 3 min in −20C methanol. Samples were washed for 5 min in PBS-T, and then incubated in blocking buffer (5% normal goat serum, 5% normal donkey serum, and 3% bovine serum albumen in PBS-T) for 15 min. Sections were then incubated with primary antibody overnight at 4C (for anti-K5/K14), or for 15 min at room temperature (all other antibodies). After washing with PBS-T, sections were incubated with secondary antibody for 10 min at room temperature. Sections were washed again and then mounted in 90% glycerol in PBS with 2.5 mg/mL p-Phenylenediamine (Sigma-Aldrich). Antibodies used were: rat anti-β4-integrin (BD Biosciences), rabbit anti-pHH3 (Cell Signaling Technology), rat anti-BrdU and rabbit anti-NuMA (Abcam), chicken anti-Keratin5/Keratin14 (Lechler lab), rabbit anti-Keratin 10 (Covance). Images were collected using a Zeiss Axio Imager Z1 fluorescence microscope with Apotome attachment. Adobe Photoshop and ImageJ software were used to process images.

## Measurement of spindle angles

Spindle angles were measured in Fiji and defined by the angle formed by a line drawn through the two spindle poles of late metaphase or anaphase spindles and a second line across the underlying basement membrane (see *Figure 1—figure supplement 1*). When spindle poles were not in the same plane, three dimensional reconstructions were generated and spindle angles were measured in rotated images.

## Hematoxylin and eosin staining

Tissue sections were fixed for 10 min in 10% PFA in distilled water. Slides were then submerged in Mayer's Hemotoxylin (Sigma-Aldrich) for 10 min, and then allowed to rinse in water for approximately 10 min. Slides were dipped 5–10 times in 10% Eosin in ethanol, then washed by dipping 10 times in 50% ethanol, 10 times in 70% ethanol, incubated 30 s in 95% ethanol, and 1 min in 100% ethanol. Slides were then dipped in xylene, allowed to air dry, and mounted in Permount (Sigma-Aldrich).

## Statistics

For all spindle orientations we used the Kolomogrov-Smirnov test. For comparison of two samples we used Student's t-test and ANOVA analysis was first performed when three or more samples were analyzed.

## Acknowledgements

We thank Don Cleveland, Elaine Fuchs and Steve Pruitt for mouse strains used in this study and members of the Lechler Lab for comments on the manuscript. This work was supported by grants R01-AR067203 and R01-AR055926 from NIAMS/NIH.

## Additional information

### Funding

| Funder | Grant reference number | Author |
|---|---|---|
| National Institute of Arthritis and Musculoskeletal and Skin Diseases | R01AR067203 | Terry Lechler |
| National Institute of Arthritis and Musculoskeletal and Skin Diseases | R01AR055926 | Terry Lechler |

The funders had no role in study design, data collection and interpretation, or the decision to submit the work for publication.

### Author contributions

Angel Morrow, Lindsey Seldin, Data curation, Formal analysis, Investigation, Methodology, Writing—review and editing; Julie Underwood, Data curation, Investigation, Methodology, Writing—review and editing; Taylor Hinnant, Data curation, Investigation, Methodology; Terry Lechler, Conceptualization, Data curation, Formal analysis, Supervision, Funding acquisition, Writing—original draft, Project administration, Writing—review and editing

### Author ORCIDs

Lindsey Seldin ⓘ https://orcid.org/0000-0002-4995-1152
Taylor Hinnant ⓘ https://orcid.org/0000-0002-8912-6851
Terry Lechler ⓘ https://orcid.org/0000-0003-3901-7013

## Ethics

Animal experimentation: This study was performed in accordance with recommendation in the Guide for the Care and Use fo Laboratory Animals of the NIH. Animals were handled according to an approved IACUC protocol (A092-18-04) from Duke University.

## Decision letter and Author response

Decision letter https://doi.org/10.7554/eLife.48482.012
Author response https://doi.org/10.7554/eLife.48482.013

## Additional files

### Supplementary files

• Transparent reporting form
DOI: https://doi.org/10.7554/eLife.48482.010

### Data availability

There are no large datasets associated with this manuscript.

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
