## [Decision Letter]

Thank you for submitting your article "Regulated spindle orientation buffers tissue growth in the epidermis" for consideration by *eLife*. Your article has been reviewed by three peer reviewers, one of whom is a member of our Board of Reviewing Editors, and the evaluation has been overseen by Anna Akhmanova as the Senior Editor. The reviewers have opted to remain anonymous.

The reviewers have discussed the reviews with one another and the Reviewing Editor has drafted this decision to help you prepare a revised submission.

Summary:

The short report by Morrow et al. sheds light on a highly interesting question of how the epidermis maintains its homeostasis and remains morphologically normal despite a large amount of oncogenic mutations. They report that this is achieved by regulation of spindle orientation (parallel or perpendicular) in dividing cells. While in adult murine back the spindle is primarily oriented parallel to the basal membrane, there is a shift towards perpendicular orientation and thus asymmetric division in embryonic skin and the footpad, both of which display a higher proliferation rate compared to adult back skin.

Morrow et al. further demonstrate that Cdkn1b expression (in embryonic epidermis) reduces, whereas TPA-treatment or oncogenic KRAS induces perpendicular spindle orientation/asymmetric cell division, and that NuMA is a key regulator.

Overall, the manuscript is conceptually interesting and provides insights into how tissue is maintained in homeostasis or upon pharmacological or genetic insults. Although it would have been valuable to know more on the mechanism of the orientation, it is a short report and future studies will further clarify the mechanism. There are few concerns that should be addressed prior publication.

Essential revisions:

1) The authors use Cdkn1b overexpressing embryonic epidermis to demonstrate that hyperproliferation induces perpendicular-cell division (Figure 1A-C). Does epidermal Cdkn1b expression also affect spindle orientation and proliferation in adult skin (back and footpad)? As it is well known that embryonic and adult skin are differently regulated in homeostasis, they should show the data in adult skin (TPA-treat back skin and/or footpad skin) using the mouse line (K14-rtTA;TRE-Cdkn1b) to show that hyperproliferation induces perpendicular cell division in adult epidermis.

2) The authors mention that NuMA mutation does not affect adult epidermis morphology. Is this true for both back skin and footpad epidermis? How does NuMA mutation alone affect the division rate and orientation of cells in the footpad?

3) Figure 3I: Surprisingly, oncogenic KRas seems to reduce proliferation of basal cells. What is the explanation for that? Is there increased proliferation in suprabasal layers? How does oncogenic KRas induce perpendicular cell division without hyperproliferation? Does it suppress delamination? The authors mention that differentiation is suppressed. However, as stated before, altered spindle orientation correlated with increased proliferation. It would be better to provide the data showing the delamination rate in WT, KRAS^G12D^, and KRAS^G12D^/NuMA^ΔMTBD^ mice. Furthermore, Authors should also show quantitation of proliferation in KRAS^G12D^/NuMA^ΔMTBD^ skin.

4) How do the authors reconcile the work from Lim et al. (Science, 2014) showing that the mechanism of probabilistic fate and neutral clonal competition rather than asymmetric cell division dominates the footpad skin?

---

## [Author Response]

Essential revisions:1) The authors use Cdkn1b overexpressing embryonic epidermis to demonstrate that hyperproliferation induces perpendicular-cell division (Figure 1A-C). Does epidermal Cdkn1b expression also affect spindle orientation and proliferation in adult skin (back and footpad)? As it is well known that embryonic and adult skin are differently regulated in homeostasis, they should show the data in adult skin (TPA-treat back skin and/or footpad skin) using the mouse line (K14-rtTA;TRE-Cdkn1b) to show that hyperproliferation induces perpendicular cell division in adult epidermis.

We quantitated spindle orientation in paw skin after Cdkn1b expression (Krt14- rtTA;TRE-Cdkn1b) and found that, like in the embryonic epidermis, there was a switch to more planar cell divisions (Figure 1—figure supplement 2). We did not perform this experiment in backskin because of the very low levels of proliferation upon Cdkn1b expression and due to the fact that spindle orientation is largely planar under control conditions.

We also attempted the suggested experiment – to determine whether Cdkn1b expression in the basal epidermis of the adult is able to reverse the effects of TPA on spindle orientation. However, the combination of TPA treatment and Cdkn1b expression resulted in dramatic effects on the skin. There was thickening of the skin and massive cellularization of the dermis, likely an inflammatory response. There were also epidermal integrity defects (cell-cell separations). Due to these findings, we did not quantitate division orientation.

**Author response image 1. respfig1:** Effect of combined TPA treatment and Cdkn1b expression on the epidermis. Top – Basement membrane (red) and nuclei (blue) staining of control, TPA-treated, and TPA-treated/Cdkn1b-expressing epidermis. Note the increased thickness of the epidermis with TPA treatment and the dramatic changes in tissue architecture when in TPA + Cdkn1b. Bottom – Image showing the epidermal integrity defects in the TPA + Cdkn1b treated skin (basement membrane is red, krt14 is green, and nuclei are blue).

2) The authors mention that NuMA mutation does not affect adult epidermis morphology. Is this true for both back skin and footpad epidermis? How does NuMA mutation alone affect the division rate and orientation of cells in the footpad?

We have included the data on division rate (Figure 3I) and spindle orientation (Figure 2—figure supplement 1). The division rate is not statistically different from controls. There is a skew to more random divisions in the NuMA mutant, as expected. Because these parameters are unaltered and the architecture looks normal, we have not included histological images in the manuscript.

3) Figure 3I: Surprisingly, oncogenic KRas seems to reduce proliferation of basal cells. What is the explanation for that? Is there increased proliferation in suprabasal layers? How does oncogenic KRas induce perpendicular cell division without hyperproliferation? Does it suppress delamination? The authors mention that differentiation is suppressed. However, as stated before, altered spindle orientation correlated with increased proliferation. It would be better to provide the data showing the delamination rate in WT, KRAS^G12D^, and KRAS^G12D^/NuMA^ΔMTBD^ mice. Furthermore, Authors should also show quantitation of proliferation in KRAS^G12D^/NuMA^ΔMTBD^ skin.

The change in proliferation rate in the KRas did not reach statistical significance in our first experiments. We repeated these results in combination with the NuMA mutant proliferation data (requested and discussed above) and again, we found no statistically significant difference (Figure 3I). We did not note any change in suprabasal proliferation.

There are several possibilities for how KRas alters division orientation. First, we cannot rule out a direct effect of KRas on division orientation. However, based on the decrease in differentiation/delamination that we observed, one possibility is that the epidermis responds to this homeostatic perturbation by increasing perpendicular divisions.

We have now included the proliferation data for the KRas/NuMA double mutant as suggested (Figure 3I). There was no significant change from control, Kras, or NuMA alone. We did not perform the suggested experiment using the K10-rtTA;TRE-H2b-GFP system to analyze differentiation as this would have required multiple rounds of matings that was not possible in the revision timeline.

4) How do the authors reconcile the work from Lim et al. (Science, 2014) showing that the mechanism of probabilistic fate and neutral clonal competition rather than asymmetric cell division dominates the footpad skin?

We do not view these works as being contradictory. A cell that undergoes asymmetric divisions and differentiation can also undergo neutral clone competition. It is an interesting question whether the kinetics of clone competition would change when you alter delamination/ACD rates in a tissue, but it is beyond the scope of this manuscript to address this.